# Towards an argumentation-based approach to explainable planning

**Anna Collins, Daniele Magazzeni** and **Simon Parsons**
Department of Informatics, King's College London
{anna.collins,daniele.magazzeni,simon.parsons}@kcl.ac.uk

## Abstract

Providing transparency of AI planning systems is crucial for their success in practical applications. In order to create a transparent system, a user must be able to query it for explanations about its outputs. We argue that a key underlying principle for this is the use of causality within a planning model, and that argumentation frameworks provide an intuitive representation of such causality. In this paper, we discuss how argumentation can aid in extracting causalities from plans and models, and how they can create explanations from them.

## 1 Introduction

Explainability of AI decision-making is crucial for increasing trust in AI systems, efficiency in human-AI teaming, and enabling better implementation into real-world settings. Explainable AI Planning (XAIP) is a field that involves explaining AI planning systems to a user. Approaches to this problem include explaining planner decision-making processes as well as forming explanations from the models. Past work on model-based explanations includes an iterative approach (Smith 2012) as well as using explanations for more intuitive communication with the user (Fox, Long, and Magazzeni 2017). With respect to human-AI teaming, the more helpful and illustrative the explanations, the better the performance of the system overall.

Research into the types of questions and motivations a user might have includes work with *contrastive questions* (Miller 2018). These questions are structured as *'Why F rather than G?'*, where $F$ is some part (i.e. action(s) in a plan) of the original solution and $G$ is something the user imagines to be better. While contrastive questions are useful, they do not consider the case when a user doesn't have something else in mind (i.e. $G$) or has a more general question about the model. This includes the scenario in which the user's understanding of the model is incomplete or inaccurate. Research in the area of model reconciliation attempts to address this knowledge gap (Chakraborti et al. 2017).

More broadly, questions such as *'Why A?'*, where $A$ is an action in the plan, or *'How G?'*, where $G$ is a (sub)goal, must be answerable and explainable. Questions like these are inherently based upon definitions held in the domain related to a particular problem and solution. The user's motivation behind such questions can vary: he could think the action is unnecessary, be unsure as to its effects, or think there is a better option. Furthermore, questions regarding particular state information may arise, such as *'Why A here?'* and *'Why can't A go here?'*. For these, explanations that include relevant state information would vastly improve their efficiency when communicating with a user (Miller 2018). This is especially true for long plans, when a user does not have access to a domain, or the domain is too complex to be easily understood. Thus, extracting relevant information about action-state causality from the model is required.

In the space of planning, causality underpins a variety of research areas including determining plan complexity (Giménez and Jonsson 2008) and heuristics (Helmert 2004). Many planners also can create causal graph visualizations of plans for a user to interact with (Pearl 2014). The general structure of causality in planning is 'action causes state'. Indirectly, this can be seen as 'action enables action', where the intermediary state is sufficient for the second action to occur. Hilton describes different 'causal chains' which mirror the types of causality found in planning; action-state causality can be identified as either a 'temporal' or 'unfolding' chain, while action-action causality is similar to an 'opportunity chain'(Hilton, McClure, and Slugoski 2005). For now, we will focus on these two types of general causality.

To represent the causality of a model, argumentation is a good candidate; as detailed by (Bochman 2005), argumentation frameworks and causal models can be viewed as two versions of one entity. A recent related work uses argumentation for explainable scheduling (Cyras et al. 2019). We consider an ASPIC$^+$ (Modgil and Prakken 2013) style framework with defeasible rules capturing the relationships between actions in a plan and strict rules capturing action-state causality. This structure allows more than a causal representation of a plan; it allows multiple types of causality to be distinguished and different causal 'chunks' to be created and combined to be used as justification for explanations.

In this paper we present an initial approach for using argumentation to represent causality, which can then be used to form more robust explanations. In the following sections, a motivating scenario will be introduced and used to showcase our current approaches of abstracting causalities and state information into argumentation frameworks.

## 2 Motivating Example

Consider a simple logistics scenario in which three trucks are tasked with delivering three packages to different locations. The user analyzing the planner output has the plan as well as a general, non-technical understanding of the model and the goals of the problem; the user knows that trucks can move between certain waypoints that have connecting roads of differing lengths, there are refueling stations at waypoints $B$ and $E$, and some subgoals of the problem are to have $package\,1$ delivered to $waypoint\,C$, $package\,2$ delivered to $waypoint\,G$, and $package\,3$ delivered to $waypoint\,D$. The user is also aware that the three trucks and three packages are at $waypoint\,A$ in the initial state. A basic map of the domain and plan are shown in Figures 1 and 2, respectively.

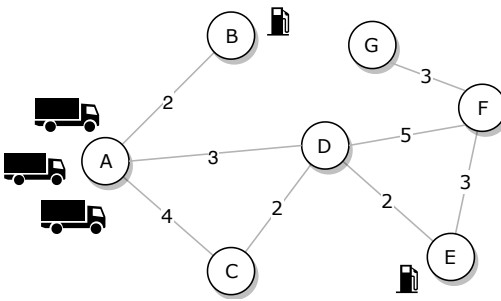

Figure 1: Example Domain Map

```
0.000: (load_truck t1 p1)      [1.000]
0.000: (load_truck t2 p3)      [1.000]
0.000: (drive_truck t3 wpB)    [2.000]
1.001: (load_truck t1 p2)      [1.000]
1.001: (drive_truck t2 wpD)    [3.000]
2.001: (drive_truck t1 wpC)    [4.000]
2.001: (refuel_truck t3)       [5.000]
4.002: (unload_truck t2 p3)    [1.000]
5.002: (unload_truck t1 p1)    [1.000]
6.003: (drive_truck t1 wpD)    [2.000]
8.004: (drive_truck t1 wpE)    [2.000]
10.005: (refuel_truck t1)      [5.000]
15.006: (drive_truck t1 wpF)   [3.000]
18.007: (drive_truck t1 wpG)   [3.000]
21.008: (unload_truck t1 p2)   [1.000]
```

Figure 2: Example Plan

Even with a simple and intuitive problem such as this, questions may arise which cannot be answered trivially. One such question is *'Why drive truck 1 to waypoint E?'*. Addressing this question requires the causal consequence of applying the action; in other words, how does driving *truck 1* to *waypoint E* help in achieving the goal(s)?

As discussed previously, tracking state information throughout a plan can be useful for explanations. This is especially true when values of state variables are not obvious at any given point in a plan and their relevance to a question is not known. A question such as *'Why drive truck 3 to waypoint B?'* has this property. These two questions will be addressed in the following sections.

## 3 Background on Argumentation

As mentioned above, in this paper we will make use of ASPIC$^+$ as the underlying argumentation system from which explanations are constructed. However, what we are suggesting is not limited to ASPIC$^+$; we can imagine using most formal argumentation systems to reason in this way. For a full description of ASPIC$^+$ see (Modgil and Prakken 2013). In this paper we only make use of the ability to construct arguments, and so that is the only aspect of the system that we describe.

We start with a language $\mathcal{L}$, closed under negation. A reasoner is then equipped with a set $\mathtt{Rules}$ of strict rules, denoted $\phi_1, \ldots, \phi_n \rightarrow \phi$, and defeasible rules, denoted $\phi_1, \ldots, \phi_n \Rightarrow \phi$, where $\phi_1, \ldots, \phi_n, \phi$ are all elements of $\mathcal{L}$. A knowledge base $\Delta$ is then a set of elements $K$ from $\mathcal{L}$ and a set $\mathtt{Rules}$. From $\Delta$ it is possible to construct a set of arguments $\mathcal{A}(\Delta)$, where an argument $A$ is made up of some subset of $K$, along with a sequence of rules, that lead to a conclusion. Given this, $\mathtt{Prem}(\cdot)$ returns all the premises, $\mathtt{Conc}(\cdot)$ returns the conclusion and $\mathtt{TopRule}(\cdot)$ returns the last rule in the argument. An argument $A$ is then:

- $\phi$ if $\phi \in K$ with: $\mathtt{Prem}(A) = \{\phi\}$; $\mathtt{Conc}(A) = \phi$; $\mathtt{Sub}(A) = \{A\}$; and $\mathtt{TopRule}(A) =$ undefined.

- $A_1, \ldots, A_n \rightarrow \phi$ if $A_i$, $1 \leq i \leq n$, are arguments and there exists a strict rule of the form $\mathtt{Conc}(A_1), \ldots, \mathtt{Conc}(A_n) \rightarrow \phi$ in $\mathtt{Rules}$. $\mathtt{Prem}(A) = \mathtt{Prem}(A_1) \cup \ldots \cup \mathtt{Prem}(A_n)$; $\mathtt{Conc}(A) = \phi$; and $\mathtt{TopRule}(A) = \mathtt{Conc}(A_1), \ldots, \mathtt{Conc}(A_n) \rightarrow \phi$.

- $A_1, \ldots, A_n \Rightarrow \phi$ if $A_i$, $1 \leq i \leq n$, are arguments and there exists a defeasible rule of the form $\mathtt{Conc}(A_1), \ldots, \mathtt{Conc}(A_n) \Rightarrow \phi$ in $\mathtt{Rules}$. $\mathtt{Prem}(A) = \mathtt{Prem}(A_1) \cup \ldots \cup \mathtt{Prem}(A_n)$; $\mathtt{Conc}(A) = \phi$; and $\mathtt{TopRule}(A) = \mathtt{Conc}(A_1), \ldots, \mathtt{Conc}(A_n) \Rightarrow \phi$.

Then, given $K = \{a; b\}$ and $\mathtt{Rules} = \{a \rightarrow c; b, c \Rightarrow d\}$, we have the following arguments:

$$A_1 : a$$
$$A_2 : b$$
$$A_3 : A_1 \rightarrow c$$
$$A_4 : A_2, A_3 \Rightarrow d$$

When applied to planning, these arguments define a subsection of a causal chain, as will be described below.

## 4 Tracing Causality

In order to utilize causality in explanations, the causal links between actions in a plan need to be extracted and abstracted into a framework. This process is planner-independent, so it requires only the plan, problem, and domain as inputs. An algorithm is used to extract the causalities which then form a knowledge base of causal links. This can then be used by an argumentation engine to construct arguments representing the causal 'chunks' in a plan. From this, questions of the forms *'Why A?'* and *'How G?'* can be addressed. This process is described in the following sections.

## 4.1 Extracting causalities from a plan

To extract causal relationships between actions in a plan, an algorithm similar to the one used in (Chrpa and Barták 2008) for detecting action dependencies is utilized:

1. Finds connections between one action's effects and another's preconditions from the domain to form a knowledge base. In general terms we can think of these chunks as being statements in some logical language of the form:

$$a \Rightarrow b$$
$$b, c \Rightarrow d$$

which denote the statements '$a$ enables $b$' and '$b$ and $c$ together enable $d$' where $a, b, c, d$ are actions in a plan.

2. Finds the subgoals, if any, that are satisfied by these causal links

Thus, part of our logistics example could be translated into the causal knowledge base:

$$((load\ truck\ t1\ p1),$$
$$(drive\ truck\ t1\ wpC)) \Rightarrow (unload\ truck\ t1\ p1)$$
$$(drive\ truck\ t1\ wpC) \Rightarrow (drive\ truck\ t1\ wpD)$$
$$(unload\ truck\ t1\ p1) \Rightarrow p1\ at\ wpC$$
$$(drive\ truck\ t1\ wpD) \Rightarrow (drive\ truck\ t1\ wpE)$$

## 4.2 Forming arguments

Given a knowledge base, the argumentation engine can construct a sequence of arguments with defeasible rules:

$$A_1 : (load\ truck\ t1\ p1)$$
$$A_2 : (drive\ truck\ t1\ wpC)$$
$$A_3 : A_1, A_2 \Rightarrow (unload\ truck\ t1\ p1)$$
$$A_4 : A_3 \Rightarrow p1\ at\ wpC$$
$$A_5 : A_2 \Rightarrow (drive\ truck\ t1\ wpD)$$
$$A_6 : A_5 \Rightarrow (drive\ truck\ t1\ wpE)$$
$$A_7 : A_6 \Rightarrow (refuel\ truck\ t1)$$
$$A_8 : A_7 \Rightarrow (drive\ truck\ t1\ wpF)$$
$$A_9 : A_8 \Rightarrow (drive\ truck\ t1\ wpG)$$
$$A_{10} : A_9 \Rightarrow (unload\ truck\ t1\ p2)$$
$$A_{11} : A_{10} \Rightarrow p2\ at\ wpG$$

These summarize the causal structure of part of the plan (i.e. a 'causal chunk' as defined in Secion 4.3), summarized in argument $A_{11}$, which can then be presented to a user who is seeking explanations. A visualization of these arguments can be seen in Figure 3.

## 4.3 Using causal chunks for explanation

We define the notion of a causal 'chunk' as any subsection(s) of the causal chain(s) extracted from the plan or model and then combined. Intuitively, these chunks can focus on one 'topic' (e.g. state variable, object instance) to provide a higher-level abstraction of causality rather than just the individual causal links. The argument $A_{11}$ which represents such a causal chunk shows only the action-action causalities

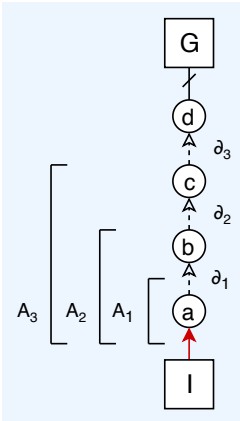

Figure 3: $I$ is the initial state, $G$ is the goal state, $a, b, c, d$ are (not necessarily sequential) actions in a plan, $\delta_i$ are the defeasible rules from the knowledge base, and $A_1, A_2, A_3$ are arguments formed from these actions and rules.

(i.e. from just one causal chain) involving the object *truck 1*. These chunks are created by searching through the Rules of the framework for those pertaining to a specific 'topic'.

Given arguments such as $A_{11}$, we propose two methods of structuring explanations. The first method is allowing the user to engage the system in a dialogue. For our example, the question, *'Why e?* where $e$ is the action of driving *truck 1* to *waypoint E* could be used to query the system:

```
why e
```

Following work such as (Parsons, Wooldridge, and Amgoud 2003), the system replies to this query by building an argument for $e$, in this case $A_6$, and using this to provide a suitable response, which might be by returning Conc($A_5$), since $A_5 \Rightarrow e$. Thus the system could reply with:

```
d, which leads to e
```

where $d$ is *drive truck t1 wpD*. The user could then continue to unpack the causal chunk by asking:

```
why d
```

and so on. This would provide the user with the causalities which enabled action $e$ to be applied. The same could be done using a forward approach where the argument $A_6$ is expanded until a subgoal is reached, if possible (e.g. $A_{11}$). The user can then ask:

```
why e
```

and the system responds with:

```
e leads to f
```

as in $A_7 : A_6 \Rightarrow f$. Iteratively, this would show how $e$ leads to some goal or subgoal. Reversing this process will also explain how a goal is reached.

The second method of structuring explanations is detailed in Section 5.2, and can be applied to this example similarly.

## 5 Extracting State Information

Using a similar method as above, causalities held within the state space of the plan are extracted and represented as a

knowledge base. An algorithm is used that iterates through the effects of actions from a plan and extracts the state variables they alter. They can then be used to answer questions such as *'Why A here?'* and *'Why can't A go here?'*. In general terms, we define these dependencies as being statements in some logical language of the form:

$$x_0, y_0, z_0$$
$$a \rightarrow \Delta x_a$$
$$b \rightarrow \Delta y_b; b \rightarrow \Delta z_b$$
$$x_f, y_f, z_f$$

which denote the statements '*a* causes $\Delta x_a$' and '*b* causes $\Delta y_c$ and $\Delta z_c$'. Here, $a, b$ are actions in the plan, and $x, y, z$ are state variables. The $x_0, y_0, z_0$ denote the values of those variables in the initial state while $x_f, y_f, z_f$ denote the final values in the goal state; $\Delta x_a$ denotes the change in $x$ after applying action $a$.

Applying this to our logistics example and the question, *'Why drive truck 3 to waypoint B?'*, these strict rules are relevant:

$$t3 \; fuel \; is \; 2$$
$$(drive \; truck \; t3 \; wpB) \rightarrow t3 \; fuel \; decrease \; 2$$
$$(refuel \; truck \; t3) \rightarrow t3 \; fuel \; increase \; 25$$
$$t3 \; fuel \; is \; 25$$

From these, it is clear the truck's fuel level is too low in the initial state to go anywhere besides waypoint B (see Figure 1). However, it is not clear why the truck does not just stay put. Alone, these rules do not provide a full explanation, but they can be added to the action-action causal chains for more complete explanations.

## 5.1 Combining different forms of causality

When used in conjunction, the causal traces and opportunity traces form a strong basis of justification for an explanation (see Figure 4 for a visual representation). Using the example from before, the relevant defeasible rules from the causal chain are:

$$(drive \; truck \; t3 \; wpB) \Rightarrow (refuel \; truck \; t3)$$
$$(refuel \; truck \; t3) \Rightarrow t3 \; fuel \; > 5$$

where the conclusion of the second rule is a subgoal of the problem, perhaps previously unknown to the user. That is, because the problem requires all trucks to have a minimum amount of fuel at the end, truck 3 had to refuel but could not deliver any packages due to its low initial fuel amount. Thus, combining arguments from both types of causal chains more aptly answers this question.

A method for seamlessly creating explanations from this structure is an intended future work. For now, it is possible to extract both the defeasible rules and strict rules governing the causal effects related to a specific topic and present them to a user. How to determine which rules are relevant to a specific user question and how to combine the rules to form higher-level causal chunks are ongoing works.

One possible method of creating relevant causal chunks is to extract *all* rules related to a specific 'topic' (e.g. state

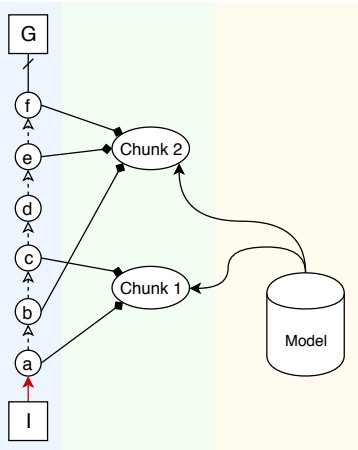

Figure 4: Rules from both the plan layer (blue) and the model layer (yellow) are combined to form the causal 'chunks'.

variable). For the variable 't3 fuel', all actions which alter it will be extracted along with any actions that enable the altering actions from the defeasible rules. Additionally, any (sub)goals containing 't3 fuel' will be extracted. Together, these form a chunk representing the causes of changes to 't3 fuel' as well as its relationship to the (sub)goals. The arguments below represent the causal 'chunk':

$A_1 : t3 \; fuel \; is \; 2$
$A_2 : A_1 \Rightarrow ((drive \; truck \; t3 \; wpB) \rightarrow t3 \; fuel \; decrease \; 2)$
$A_3 : A_2 \Rightarrow ((refuel \; truck \; t3) \rightarrow t3 \; fuel \; increase \; 25)$
$A_4 : A_3 \Rightarrow t3 \; fuel \; > 5$

where the conclusion of $A_3$ is a subgoal of the problem.

## 5.2 More forms of explanation

When unpacked iteratively, the arguments in the causal chunk centred on 't3 fuel' would give a similar output explanation as in the example in Section 4.3. For example, a user asking the question *'Why b?'* where $b$ is the action *(drive truck 3 to waypoint B)* would either receive the response:

*t3 fuel is 2* `enables` $b$

or the response:

$b$ `causes` *t3 fuel decrease 2* `and` `enables` $c$

if using a forward chaining approach, where $c$ is the premise of the conclusion of $A_2$, *(refuel truck t3)*. This process would continue until the subgoal *t3 fuel >5* is reached. However, identifying what state variables are relevant given a user question is not trivial. The question *'Why drive truck 3 to waypoint B?'* has no mention of the truck's fuel, so its relevance must be deduced from the plan, problem and domain.

Another method of providing explanations is through a graph structure, as depicted in Figure 5. Given a query, the relevant causal chunks would be identified and represented in the graph with individual actions and state changes as nodes and the causal rules between them as edges. This approach could also help explain question of the form, *Why*

*can't A go here?*, as inapplicable actions (ones not in the plan) can be shown. Developing a robust system such as this is important future work.

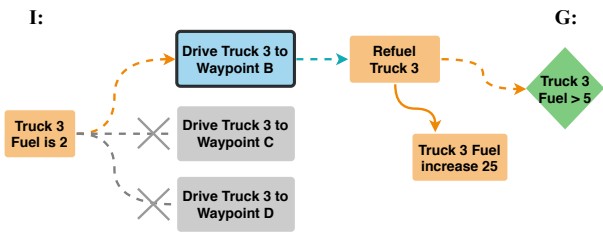

Figure 5: An example graph with the queried action in blue and nodes contained in the 't3 fuel' chunk in orange, and *I* and *G* the initial and goal states. Dashed edges denote defeasible rules; solid edges denote strict rules.

## 6 Discussion

We acknowledge that this is a preliminary step and more work is required to expand on the ideas presented in this paper. One such future work involves defining exactly what questions, which range from action-specific to model-based, can be answered and explained using our approach. Also, how these questions are captured from a user is an open question. The query, *'Why didn't truck 3 deliver any packages?'* can be answered using the causal information captured in the framework, but how one converts this question to a form that the system understands requires further research. Potential methods for communicating a user question include a dialogue system or Natural Language Processing techniques.

Along with expanding the set of questions that can be addressed, extensions to the argumentation framework itself should be considered. Better methods for creating causal 'chunks' for specific user questions are needed. It may be advantageous to use argumentation schemes to help identify relevant topics of chunks and which causal chains should be included from the framework. This relates to the idea of 'context' and identifying the motivation of a question. If the system can be more precise in extracting the relevant information, the explanations themselves will be more effective.

Related to this is the need to explore other ways of presenting an explanation to a user. Research into the efficacy of explanations and how to properly assess the effectiveness of the explanations in practice are future areas of research, and will require user studies. Our starting point will be the approach outlined in Section 4.3 which has been shown empirically to be effective in contexts such as human-robot teaming (Sklar and Azhar 2015).

## 7 Conclusion

In this paper we proposed an initial approach to explainable planning using argumentation in which causal chains are extracted from a plan and model and abstracted into an argumentation framework. Our hypothesis is that this allows ease of forming and communicating explanations to a user. Furthermore, causal 'chunks' can be created by combining rel-

evant causal links from the chains which explain the causalities surrounding one 'topic'. We believe these help with making more precise explanations, and that chunks can be used to provide hierarchical explanations. Overall, the approach is a first step towards exploiting the intuitive functionality of argumentation in order to use causality for explanations.

**Acknowledgements**  This work was partially supported by EPSRC grant EP/R033722/1, Trust in Human-Machine Partnership, and by a PhD studentship from the Faculty of Natural and Mathematical Sciences at King's College London.

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
