# OpenReview forum: "Towards an argumentation-based approach to explainable planning"
_icaps-conference.org/ICAPS/2019/Workshop/XAIP — XAIP 2019_

### Official Review · AnonReviewer1 · 2019-05-09
**Insightful paper on causality to generate explanations**

**Rating:** 3
**Confidence:** 2

**Review:**

This is a well-written paper presenting multiple methods of generating explanations using argumentation systems with a planner. The major claim of the paper is that causality is crucial is a required underlying principle for generating good explanations. The paper walks through multiple concrete examples for both a variety of questions that could be asked and how causal reasoning can answer those questions.  The application of argumentation systems with a planner for explanation is novel. This work is highly relevant to the workshop.

Question for the authors:
It seems that using an argumentation system could provide many such possible explanations. How would you go about selecting a small number (or single) explanation to offer a user?

---

### Official Review · AnonReviewer4 · 2019-05-13
**argumentation used for simple plan-causality queries**

**Rating:** 2
**Confidence:** 2

**Review:**

The paper starts out with the aim to integrate argumentation as a tool for XAIP. I applaude this motivation and think it highly desirable to pursue this direction.

I am however rather non-plussed by the paper at hand. At the end, so far as I get it, what the authors analyze here using formalisms from argumentation comes down to simple plancausality analyses as have been known in partial-order causal-link planning since eternities. In particular, various previous works have proposed to analyze such causal links for explanining to a user the inner workings of a plan. One example citation is Seegebarth et al ICAPS 2012. Inhowfar do the techniques presented here go beyond this? As far as I can tell, not much or not at all. Certainly it's not my duty as reviewer to answer this question.

I do think that a discussion of the possible uses of argumentation may be highly interesting at the workshop. But I fail to see how the paper at hand could form the basis for such a discussion. I think the paper can be accepted depending on space and the shape of the program; but I don't think it would be a big loss to leave it out.

Some notes on the write-up:

The introduction starts with a long generic text hardly specific to the paper at hand. Then argumentation is introduced curtly, not making clear what its role/perception here is. In partiular, the key terms "defeasible" and "strict" rules are not made clear. Even after reading the whole paper it is not clear to me what the difference between these is and inhowfar this is relevant to the analysis of plans. This needs to be made much clearer given the purpose of the paper addressed at a planning community. The same applies to Section 3. Instead of, or in addition to, the formal definition, there should be an intuitive explanation what these things serve for and what the difference between strict and defeasible rules is.

Sec 4.3 The description of things one can do based on the formalization is underwhelming. This looks just like straight old causal link analysis to me, cf above.

I also find it hard to see more interest/novelty in the following discussions. The claim/hint from the introduction that this s "more than a causal representation of a plan" is not being substantiated/supported in a way I can appreciate. This may be just a write-up issue, but essentially I do think there is very little here in support of how the argumentation formalization is yielding a genuine advatage, rather than just introducing heavy notation for very simple analyses that could ave been formulated more simply.

---

### Official Review · AnonReviewer2 · 2019-05-13
**Just what it says on the tin**

**Rating:** 4
**Confidence:** 2

**Review:**

The paper contains exactly what I expected based on its title. It's pretty well structured and does what I expect from a "Towards" paper, although it leaves me with some questions. First, I still don't really understand why argumentation. This is because I don't have any background on the topic. The formal preliminaries given aren't helpful in describing to me why it's useful, as I don't know what problem argumentation is supposed to solve. This is related to a second problem: I'm not sure what you want your explanations to look like. In the end, I know you want a causal description in natural language, but I'm more concerned with the content rather than the syntax. I suspect that the explanation found is a "causal chunk" as discussed in 5.1, since 5.2 delves into "more" forms of explanation. However, this is never explicitly stated, and more importantly, you need to argue why this is a good explanation, i.e., will satisfy a user. You can't hand-wave the issue of what makes the explanation good as "something to be investigated later", because this is the rationale for why you think argumentation is a good direction to explore. If you aren't aiming for a useful solution, there's no point in doing the work to get there.

Overall, this seems like a good paper to discuss at a workshop.

Notes:
Section 4.3 would be much easier to follow if you stuck with the example and used actions rather than letters with no context.
Figure 4 confuses rather than illuminates. What's X? What do the colors signify? What do a-d label, and how are they related to A1-3 (from section 4.2? or section 5.1?).
The purpose of section 5.2 is mysterious to me. At the moment, I interpret it as saying, "Hey, if this whole argumentation thing doesn't work, here's a thumbnail sketch of an alternative." If that's so, I'd prefer you flesh out your first idea for now rather than introducing a second with little detail.

---

### Decision · Program_Chairs · 2019-05-15

**Decision:**

Accept

**Comment:**

While the reviewers do not fully agree on the decision, in the spirit of making the workshop a venue for discussion and feedback we decided to reject only those papers with strong reject votes.

Please address all review criticism as best possible for the final paper version and its presentation at the workshop. Looking forward to discuss your work at the workshop!